# Porphylipoprotein Accumulation and Porphylipoprotein Photodynamic Therapy Effects Involving Cancer Cell-Specific Cytotoxicity

**DOI:** 10.3390/ijms22147306

**Published:** 2021-07-07

**Authors:** Hiromi Kurokawa, Hiromu Ito, Hirofumi Matsui

**Affiliations:** 1Faculty of Medicine, University of Tsukuba, Ibaraki 305-8577, Japan; 2Graduate School of Medical and Dental Sciences, Kagoshima University, Kagoshima 890-8544, Japan; k8459616@kadai.jp (H.I.); hmatsui@md.tsukuba.ac.jp (H.M.)

**Keywords:** reactive oxygen species, photodynamic therapy, porphylipoproteins, cholangiocarcinoma

## Abstract

In photodynamic therapy (PDT) for neoplasms, photosensitizers selectively accumulate in cancer tissue. Upon excitation with light of an optimal wavelength, the photosensitizer and surrounding molecules generate reactive oxygen species, resulting in cancer cell-specific cytotoxicity. Porphylipoprotein (PLP) has a porphyrin-based nanostructure. The porphyrin moiety of PLP is quenched because of its structure. When PLP is disrupted, the stacked porphyrins are separated into single molecules and act as photosensitizers. Unless PLP is disrupted, there is no photosensitive disorder in normal tissues. PLP can attenuate the photosensitive disorder compared with other photosensitizers and is ideal for use as a photosensitizer. However, the efficacy of PLP has not yet been evaluated. In this study, the mechanism of cancer cell-specific accumulation of PLP and its cytotoxic effect on cholangiocarcinoma cells were evaluated. The effects were investigated on normal and cancer-like mutant cells. The cytotoxicity effect of PLP PDT in cancer cells was significantly stronger than in normal cells. In addition, reactive oxygen species regulated intracellular PLP accumulation. The cytotoxic effects were also investigated using a cholangiocarcinoma cell line. The cytotoxicity of PLP PDT was significantly higher than that of laserphyrin-based PDT, a conventional type of PDT. PLP PDT could also inhibit tumor growth in vivo.

## 1. Introduction

Photodynamic therapy (PDT) for neoplasms has the following three components: a photosensitizer (PS), light, and oxygen [1]. Upon exposure to light of an optimal wavelength, the PS molecule is excited from its ground singlet state (S0) to an excited singlet state (S1). Because S1 is unstable, the PS molecule returns to S0 by dissipating the gained energy through nonradiative decay or fluorescence. In another mechanism for attaining stability, the PS molecule undergoes an intersystem crossing to the excited triplet state (T1). In this T1 state, the PS molecule can transfer its energy by phosphorescence or by colliding with other molecules to create chemically reactive species via two types of reactions. In type I reactions, the PS molecule in the T1 state reacts with a number of substrates or solvents and transfers an electron or a proton to form a radical anion or cation species, respectively [2]. In particular, the PS molecule in the T1 state reacts with oxygen to form a superoxide anion radical. In type II reactions, the PS molecule in the T1 state reacts directly with triplet oxygen via the transfer of energy to form singlet oxygen [3]. Both superoxide anion radical and singlet oxygen are reactive oxygen species (ROS). ROS are well known for inducing potent cytotoxicity. Therefore, PDT, through the generation of ROS, injures the cancer tissue [4]. PDT induces cytotoxicity in tumor tissue because of cancer-specific PS accumulation. The mechanism of PS accumulation into cancer tissue was reported. PS bind circulating cells in the blood and diffuse throughout the parenchyma of the organ or tumor to which it has been delivered [5]. Intracellular PS accumulates through the LDL receptor [6]. Hyaluronic acid enhances PS accumulation [7]. However, sometimes these phenomena do not occur and, therefore, this mechanism cannot be accepted as a general process leading to cancer-specific PS accumulation. The details of the mechanism of cancer cell-specific accumulation of PS remain unknown.

Porphyrin is a PS that chelates iron to form heme. Heme is taken up by cells through heme carrier protein 1 (HCP1) [8]. Because the structure of porphyrin is almost the same as that of heme, we hypothesized that porphyrin is taken up by cells through HCP1. We reported that the accumulation of porphyrin in cancer cells, in which HCP1 was knocked down, decreased compared with that in control cells [9]. In contrast, porphyrin accumulation increased in cells overexpressing HCP1 [9]. Moreover, unlike normal cells, which show a low expression of HCP1, three cancer cell types were shown to distinctly expresse HCP1. These results indicate that cancer cells uptake porphyrin via HCP1 [9]. From these results, porphyrin, taken up by cells via HCP1, accumulates in large amounts in cancer cells compared to normal cells.

Furthermore, we investigated the factors regulating the expression of HCP1. HCP1 expression has been reported to be affected by hypoxic conditions in which mitochondria generate ROS [10,11]. We evaluated the relationship between mitochondrial ROS (mitROS) and HCP1 expression using rat gastric mucosal cells, RGM1, its cancer-like mutants, RGK1, obtained by treatment with carcinogenic chemicals, and manganese superoxide dismutase (MnSOD)-overexpressing RGK cells, RGK-MnSOD. These cells were previously established in our laboratory [12,13,14]. HCP1 protein levels were analyzed using Western blotting. HCP1 expression levels in RGK1 cells were greater than those in RGM1 cells. In addition, the levels of HCP1 decreased in RGK-MnSOD. These results suggest that mitROS can upregulate the expression of HCP1 [15].

Recently, PS have been conjugated to carrier molecules that deliver them specifically to tumor tissues. These carrier molecules include peptides, low-density lipoproteins (LDLs), antibodies, nanoparticles, and polymers [16]. Porphylipoprotein (PLP) is a PS that contains a hydrophobic drug-loadable core, enveloped in a porphyrin–lipid monolayer, and constrained by ApoA-1 mimetic R4F (Ac-FAEKFKEAVKDYFAKFWD) peptide networks [17]. PLP is a porphyrin-based ultrasmall nanostructure mimicking natural lipoproteins that engages the biocompatible nanostructure of lipoproteins and the multifunctionalities of naturally derived porphyrin molecules. The porphyrin moiety in PLP is quenched because of its structure. When PLP is disrupted, a stack of porphyrin is separated into single molecules which can then act as a PS. Thus, until PLP is disrupted, photosensitivity disorder is not induced in normal tissues. This indicates that PLP can attenuate the photosensitivity disorder compared with other PSs. The efficacy of PLP has been demonstrated in vitro and in vivo [18,19]. However, the mechanism of cancer cell-specific accumulation of PLP is not sufficiently understood.

In this study, we evaluated whether (1) PLP accumulation is a cancer cell-specific phenomenon, (2) HCP1 is one of the PLP uptake transporters, and (3) PDT with PLP could be an efficient treatment for cholangiocarcinoma.

## 2. Materials and Methods

### 2.1. Photosensitizers

Laserphyrin was purchased from Meiji Seika Pharma (Tokyo, Japan). PLP was synthesized by Nippon Fine Chemical Co., Ltd. (Tokyo, Japan). Both these photosensitizers are water-soluble.

### 2.2. Cell Culture

RGM1, a rat gastric epithelial cell line, was purchased from RIKEN CELLBANK (Ibaraki, Japan) and cultured in DMEM/F12 with L-glutamine (Life Technologies Japan Ltd., Tokyo, Japan). RGK1, a chemically induced oncogenic cancer-like mutant of RGM1, was established in our laboratory [13]. RGK-MnSOD, which is a manganese superoxide dismutase-overexpressing RGK1 cell line, was also established in our laboratory [14]. RGK1 cells were cultured in DMEM/F12 without L-glutamine (Sigma-Aldrich Japan K.K., Tokyo, Japan). The human gall bladder carcinoma cell line NOZ was purchased from the Japan Research BioSource Cell Bank (Osaka, Japan) and cultured in William’s E medium (Wako Pure Chem. Ind. Ltd., Osaka, Japan). The culture media contained 10% inactivated fetal bovine serum (Hyclone, Thermo Fisher Scientific K.K., Tokyo, Japan) and 1% penicillin/streptomycin (Wako Pure Chem. Ind. Ltd.). All the cells were cultured in an atmosphere of 5% CO_2_ at 37 °C.

### 2.3. siRNA-Mediated Downregulation of HCP1

Rat siRNA oligos for HCP1 were purchased from Sigma-Aldrich and were transfected into RGK1 cells using the Lipofectamine^TM^ RNAiMAX transfection reagent (Thermo Fisher Scientific), according to the manufacturer’s protocol. Briefly, 500 µL of Opti-MEM™ I Reduced Serum Medium (Thermo Fisher Scientific) and 3 µL of 10 µM siRNA were added to a 35 mm cell culture dish and mixed gently. Five microliters of Lipofectamine^TM^ RNAiMAX transfection reagent was added, and the mixture was incubated for 15 min at room temperature after gentle mixing. RGK1 cells suspended in Opti-MEM™ I Reduced Serum Medium were added (6 × 10^4^ cells/mL × 2.5 mL) to the transfection mixture and incubated for 48 h in a CO_2_ incubator at 37 °C. After incubation, the cells were collected, and total RNA was isolated. Total cDNA was synthesized from the isolated RNA using ReverTra Ace^®^ qPCR RT Master Mix with gDNA Remover (Toyobo Co., Ltd., Osaka, Japan). The synthesized cDNA was used for quantitative PCR analysis to confirm siRNA transfection and knockdown of HCP1. The cDNA was mixed with PowerUP^TM^ SYBR^TM^ Green Master Mix (Thermo Fisher Scientific) and primers, and gene amplification was performed using the StepOnePlus^TM^ real-time PCR system (Thermo Fisher Scientific). The conditions used for reverse transcription and PCR were as follows: 50 °C for 2 min and 95 °C for 2 min followed by 40 cycles of denaturation at 95 °C for 15 s, annealing at 54 °C for 15 s, and elongation at 72 °C for 1 min. The levels of HCP1 were normalized to that of an internal standard control, β-actin. The primer sets for HCP1 and β-actin gene amplification were as follows:

rHCP1 forward: 5′-TGAGCTAAGCACACCCCTCT-3′

rHCP1 reverse: 5′-TCCGTACCCTGTGAACATGA-3′

β-actin forward: 5′-AGCCATGTACGTAGCCATCC-3′

β-actin reverse: 5′-CTCTCAGCTGTGGTGGTGAA-3′

### 2.4. Cell Viability Assay

Cell viability was measured using the Cell Counting Kit-8, according to the manufacturer’s protocol [20]. RGK1 and RGM1 cells were seeded in 96-well plates at a density of 1 × 10^4^ cells/well, and NOZ cells were seeded in 96-well plates at a density of 5 × 10^3^ cells/well. The cells were then incubated overnight. The supernatant was aspirated, and the culture medium was replaced. The cells were then incubated in medium containing 0, 0.125, 0.375, 1.25, 3.75, 12.5, 37.5, and 125 μM PLP or laserphyrin at 37 °C for 24 h. After incubation, the supernatant was aspirated, and the cells were further incubated with a medium containing 10% Cell Counting Kit-8. The absorbance at 450 nm was measured using a Synergy H1 microplate reader (BioTek Instruments Inc., Winooski, VT, USA).

### 2.5. Localization of PLP in the Cells

RGK1 and RGM1 cells were seeded in 4-well chambers slide at a density of 2 × 10^4^ cells/well, and NOZ cells were seeded at a density of 1 × 10^5^ cells/well. The cells were then incubated overnight. The supernatant was aspirated, and the medium was replaced with 3.75 μM PLP. The cells were further incubated at 37 °C for 0, 0.5, 1, and 3 h. After incubation, the supernatant was aspirated, and the cells were rinsed with PBS. The medium was replaced with Hanks’ balanced salt solution (Wako Pure Chem.). The intensity of intracellular fluorescence due to PLP was measured using a fluorescence microscope, IX83 (Olympus, Tokyo, Japan). The excitation wavelength was 635–675 nm, and the emitted fluorescence was collected using a 696–736 nm filter.

### 2.6. Intracellular Accumulation of PLP

RGK1, RGM1, and NOZ cells were seeded in 12-well plates at a density of 1 × 10^5^ cells/well and incubated overnight. The supernatant was aspirated, and the medium was replaced with 3.75 μM PLP. The cells were further incubated at 37 °C for 0, 0.5, 1, and 3 h. After incubation, the supernatant was aspirated, and the cells were rinsed with PBS. The cells were then lysed in 100 μL RIPA buffer. The cell homogenates were transferred to a 96-well plate. The intensity of PLP fluorescence was measured using a Synergy H1 microplate reader. The excitation and emission wavelengths were 420 and 675 nm, respectively.

### 2.7. PDT Treatment and Cell Viability Assay

RGK1 and RGM1 cells were seeded in 96-well plates at a density of 2 × 10^3^ cells/well and incubated overnight. NOZ cells were seeded in 96-well plates at a density of 5 × 10^3^ cells/well. The cells were then incubated at 37 °C for 48 h. Thereafter, the cells were incubated with 0, 3.75, and 6.25 μM PLP or laserphyrin for 3 h and then rinsed with PBS. The medium was replaced with fresh medium lacking phenol red. For PDT, the cells were irradiated with a laser light (671 nm, 3 J/cm^2^) using an ML6600 system (Modulight, Tampere, Finland). After irradiation, the cells were incubated for 24 h. The medium was replaced with fresh medium containing 10% Cell Counting Kit-8 reagent, and further incubation was done. The absorbance at 450 nm was measured using a Synergy H1 microplate reader.

### 2.8. Effect of HCP1 Expression on the Intracellular Accumulation of PLP

RGK1 cells were cultured in 24-well plates at a density of 2.5 × 10^4^ cells/well. To enhance the expression of HCP1, the cells were exposed to 1 mM IND for 1 h. After treatment, the cells were incubated in fresh medium for 24 h. Thereafter, the cells were incubated with 0 or 3.75 µM PLP for 3 h. Subsequently, the supernatant was aspirated, and the cells were rinsed with PBS. These cells were lysed in 100 μL RIPA buffer. The cell homogenates were transferred to a 96-well plate. The intensity of fluorescence emitted by PLP was measured using a Synergy H1 microplate reader. The excitation and emission wavelengths were 420 and 675 nm, respectively.

RGK1 and RGK-MnSOD cells were seeded in 24-well plates at a density of 1 × 10^4^ cells/well and incubated for 48 h. The supernatant was aspirated, and the medium was replaced with 3.75 μM PLP. The cells were further incubated at 37 °C for 0, 0.5, 1, and 3 h. After incubation, the supernatant was aspirated, and the cells were rinsed with PBS. These cells were then lysed in 100 μL RIPA buffer. The cell homogenates were transferred to a 96-well plate. The intensity of fluorescence emitted by PLP was measured using a Synergy H1 microplate reader. The excitation and emission wavelengths were 420 and 675 nm, respectively.

RGK1 cells transfected with HCP1 siRNA for 48 h were seeded in 24-well plates at a density of 4.5 × 10^4^ cells/well and incubated overnight. The medium was replaced with fresh medium containing 3.75 µM of PLP and further incubated for 0, 0.5, 1, and 3 h. After incubation, the cells were washed with PBS and lysed in 100 μL RIPA buffer. The cell lysate was transferred to a 96-well plate, and the fluorescence was measured using an Infinite M200 microplate reader (Tecan Japan Co., Ltd., Kawasaki, Japan), with excitation and emission wavelengths of 420 and 675 nm, respectively.

### 2.9. PDT Treatment of Tumor-Bearing Mice

BALB/c nude mice were obtained from Oriental Yeast (Tokyo, Japan). The mice were bred under conventional conditions, housed in plastic cages, and allowed free access to food and water with a light–dark cycle of 12:12 h. The animals were divided into three groups: tumor control, laserphyrin PDT, and PLP PDT. The mice were placed under general anesthesia with isoflurane, and a total of 2 × 10^6^ NOZ cells suspended in 20 µL of medium were injected into their right paw. When the tumor volume reached approximately 60 mm, 8 mg/kg laserphyrin or 5 mg/kg PLP was administered intravenously. Three hours after the administration of laserphyrin or 24 h after the administration of PLP, these animals were subjected to laser treatment. Each animal was anesthetized with isoflurane. The tumor region of animals under anesthesia was irradiated with a laser light (671 nm, 75 J/cm^2^) using the ML6600 system (Modulight, Tampere, Finland) for PDT. After irradiation, the tumor size was measured using a caliper.

## 3. Results

### 3.1. Cytotoxicity of PLP and Laserphyrin in Cancer and Normal Cells

The cytotoxicity of PLP and laserphyrin was measured using the WST assay. The viability of RGK1 cells treated with 125 µM PLP was significantly lower compared with that of untreated cells (0 µM PLP; Figure 1a). Moreover, laserphyrin was cytotoxic at 37.5 µM or higher concentrations (Figure 1b). PLP and laserphyrin did not show acute toxicity in RGM1 cells (Figure 1c,d). Based on these results, we used a concentration less than or equal to 12.5 µM.

### 3.2. Intracellular PLP Accumulation Is Accelerated in Cancer Cells Compared with Normal Cells

Both RGK1 and RGM1 cells were incubated in culture medium containing PLP, and the intensity of intracellular fluorescence was measured using a fluorescence microscope. The fluorescence intensity of PLP in cancer cells was observed starting 30 min after incubation and increased in a time-dependent manner. In normal cells, the fluorescence was observed after almost 3 h of incubation (Figure 2b). After PLP treatment, the cells were lysed with RIPA buffer. Intracellular PLP accumulation was measured using a microplate reader. PLP accumulation in cancer cells was significantly greater at all time points compared with that in normal cells (Figure 2c).

### 3.3. PLP PDT Is More Cytotoxic Than Laserphyrin PDT

The cytotoxicity of PDT using PLP or laserphyrin was determined. The viability of normal cells treated with 3.75 μM PLP PDT was 79.0% ± 1.68%, whereas that of cancer cells was 40.5% ± 4.68%. In contrast, the viability of normal cells treated with 6.25 μM PLP PDT was 38.4% ± 3.38%, and that of cancer cells was 14.0% ± 1.33% (Figure 3a). After both treatments, the viability of cancer cells was significantly decreased compared with that of normal cells, suggesting that the cytotoxicity of PLP PDT is cancer cell-specific. Laserphyrin PDT was not cytotoxic to normal and cancer cells at 3.75 and 6.25 μM (Figure 3b). Thus, compared with laserphyrin, PLP induced cytotoxicity at low concentrations.

### 3.4. Relationship between PLP Uptake and HCP1 Expression

We previously reported that intracellular accumulation of porphyrin increased via uptake through the HCP1 transporter. We asked whether the intracellular accumulation of PLP correlated with HCP1 expression in several cell lines. We reported that the expression of HCP1 was increased by indomethacin (IND) in RGK1 cells via an increase in ROS production [21]. RGK1 cells were incubated with or without IND for 1 h, and then intracellular fluorescence due to the accumulation of PLP was measured. Fluorescence intensity in RGK1 cells treated with IND was significantly higher than that in untreated cells (Figure 4a). The expression of HCP1 in RGK-MnSOD cells was lower than that in RGK1 cells. Intracellular ROS production in RGK-MnSOD was lower than in RGK1. Compared with RGK-MnSOD cells, fluorescence intensity in RGK1 cells was higher (Figure 4b). Moreover, we prepared HCP1-knockdown RGK1 cells and examined the fluorescence intensity of the accumulated PLP. The fluorescence intensity in HCP1-knockdown cells was significantly lower than that in RGK1 cells (Figure 4c).

### 3.5. Cytotoxicity of PLP in NOZ Cells

PLP did not show acute toxicity in NOZ cells at concentrations below 37.5 μM. Cell viability was 49% at 125 μM and was significantly lower compared to that at 0 μM (Figure 5).

### 3.6. Intracellular PLP Accumulation in NOZ Cells

The accumulation of PLP in NOZ cells was evaluated. Fluorescence intensity of PLP was detected 30 min after the treatment of NOZ cells, similar to what observed in RGK1 cells, and it became stronger in a time-dependent manner (Figure 6a). The PLP accumulation in NOZ cells was also measured using a microplate reader. After PLP treatment, the cells were lysed in RIPA buffer. The fluorescence intensity of PLP was measured and normalized with respect to the protein content. The fluorescence intensity of PLP in NOZ cells increased in a time-dependent manner (Figure 6b).

### 3.7. Cytotoxic Effect of PLP and Laserphyrin PDT on NOZ Cells

The cytotoxicity of PLP and laserphyrin PDT in NOZ cells was investigated. The viability of cells subjected to 3.75 μM PLP PDT or laserphyrin PDT was 65% ± 6.24% and 88.9% ± 5.52%, respectively, whereas that of cells subjected to 6.25 μM PLP PDT or laserphyrin PDT was 26.5% ± 6.04% and 77.5% ± 2.44%, respectively (Figure 7). At both concentrations, the viability of cells subjected to PLP PDT was significantly lower compared with that of cells subjected to laserphyrin PDT.

### 3.8. Inhibitory Effects of PLP PDT and Laserphyrin PDT on the Growth of NOZ Tumors

The cytotoxicity of PDT with PLP and laserphyrin was evaluated in vivo. Tumor growth in mice was observed after irradiation. Tumor volumes were measured using two orthogonal measurements, length and width, using the following formula: volume = length × width2/2 [22,23]. Tumor growth was observed to be aggressive, and the tumor volume showed a time-dependent increase. On day 25 of PLP PDT and laserphyrin PDT, the growth of the tumors was significantly suppressed compared with that in the control group (Figure 8). Considering the tumor volume in the control to be 100%, the volumes in PLP PDT and laserphyrin-PDT groups were 24% and 6%, respectively.

## 4. Discussion

In this study, we demonstrate, for the first time, that intracellular PLP accumulation in cancer cells is significantly higher than that in normal cells. PLP accumulation is related to the expression of HCP1. Moreover, the cytotoxicity of PDT using PLP was found to be effective for treating bile duct cancer. After 30 min of incubation, the fluorescence of PLP in cancer cells was observed under a fluorescence microscope, and its intensity was found to increase in a time-dependent manner (Figure 2a), whereas in normal cells, fluorescence was observed after 3 h of incubation (Figure 2b). In comparison with normal cells, the intracellular accumulation of PLP was significantly higher in cancer cells at all time points (Figure 2c). Zheng et al. reported that the fluorescence of PLP was associated with tumor tissue [19]. However, PLP accumulation is also accelerated in the liver, kidney, and lung. These results did not confirm the cancer cell-specific accumulation of PLP. We evaluated cancer cell-specific accumulation of PLP using RGM1 and RGK1 cells, the latter being cancer-like mutant cells induced by treatment of RGM1 with oncogenic chemicals. Thus, the genetic makeup of RGK1 and RGM1 cells is basically the same. The amount of PLP accumulated in RGM1 cells was significantly greater than that in RGM1 cells. Thus, the uptake of PLP is a cancer cell-specific phenomenon. We compared the effect of PLP PDT with that of laserphyrin PDT. Laserphyrin PDT has already been used in clinical applications. PLP PDT induced cancer cell-specific cytotoxicity at concentrations of 3.75 µM and higher, and the viability of the cells was decreased in a dose-dependent manner. PLP PDT was cytotoxic at lower concentrations than laserphyrin PDT. Laserphyrin PDT induced photosensitivity as a side effect. PLP PDT can achieve cancer cell-specific cytotoxicity and attenuation of side effects.

Next, we investigated the mechanism underlying the uptake of PLP. We previously reported that accumulation of porphyrin increased in cells expressing high levels of HCP1. Because the structure of chlorin, which is present in PLP, is similar to that of porphyrin, we hypothesized that PLP would be taken up through HCP1. We evaluated the relationship between PLP accumulation and HCP1 expression. Exposure of cells to IND, a nonsteroidal anti-inflammatory drug, increases the expression of HCP1 [21]. RGK1 cells were exposed to 1 mM IND for 1 h, and the amount of PLP accumulated in the cells was measured. Compared with untreated cells, PLP accumulation was significantly increased in IND-treated cells (Figure 4a). We isolated MnSOD-overexpressing cells, RGK-MnSOD. The expression of HCP1 in RGK-MnSOD cells was lower than that in RGK1 cells [15]. The accumulation of PLP in RGK-MnSOD was significantly lower than that in RGK1 cells (Figure 4b). From these results, HCP1 expression, which is regulated by intracellular ROS production, is positively related to PLP uptake. Moreover, we isolated HCP1-knockdown cells to confirm the role of HCP1 in PLP accumulation. The accumulation of PLP in HCP1-knockdown cells was significantly lower than that in RGK1 cells (Figure 4c). Based on these results, we considered two hypotheses: (i) PLP accumulated through HCP1, and (ii) chlorin was released from PLP upon its disruption and accumulated through HCP1. In addition, the intensity of fluorescence derived from PLP was increased upon upregulation of HCP1 and was decreased in cells with low expression or downregulation of HCP1.

Biliary tract cancer arises from the biliary epithelium of the small ducts in the periphery of the liver and the main ducts of the hilum [24,25]. The classification of this type of cancer is based on the anatomical location and includes intrahepatic, perihilar, and distal cholangiocarcinoma. Surgery is the main treatment for this cancer; however, resection of the intrahepatic bile ducts has limitations and is highly invasive. Moreover, residual cancer may be observed in the bile duct stump on postoperative histopathological examination, and postoperative chemotherapy may be required [26]. The chemotherapy regimen of gemcitabine and cisplatin is often used, but conclusive evidence for its efficacy is lacking. Thus, a more effective and safe treatment is required. PDT is an effective treatment for biliary tract cancers [27,28,29,30]. Ortner et al. reported that stenting and subsequent PDT treatment resulted in prolonged survival compared to stenting alone [31]. Wagner et al. reported that PDT improves palliation and survival in patients with nonresectable hilar bile duct cancer [32]. In this study, we investigated the effect of PLP PDT on the NOZ cell line, which represents a human carcinoma of the gallbladder and extrahepatic biliary tract. Accumulation of PLP in NOZ cells was increased in a time-dependent manner, similar to that in RGK1 cells (Figure 6). Both PLP PDT and laserphyrin PDT significantly decreased cell viability in a dose-dependent manner (Figure 7). Compared to laserphyrin PDT, cell viability was significantly decreased in PLP PDT at 3.75 and 6.25 µM. These results indicated that the cytotoxicity of PLP PDT was strong compared with that of laserphyrin PDT at these concentrations. We evaluated the effect of PLP PDT on NOZ tumor-bearing mice. The conditions of PLP PDT and laserphyrin PDT were determined based on previous studies [17,33]. Laserphyrin PDT was used as a positive control. We demonstrated in preliminary experiments that 4 mg/kg of laserphyrin PDT did not result in tumor regression. Thus, in this study, we used 8 mg/mL of laserphyrin PDT. Tumor size was measured sequentially by calipers, and the tumor volume was calculated. The tumor volume in control mice increased in a time-dependent manner. Both PLP PDT and laserphyrin PDT inhibited tumor progression. Compared with the control, tumor volume after PLP PDT was significantly smaller on days 13 and 25, whereas after laserphyrin PDT, it was smaller on days 10, 13, and 25 (Figure 8). Compared with laserphyrin PDT, the cytotoxicity induced by PLP PDT was significantly high at same concentration in vitro (Figure 7). In in vivo assays, the concentration of PLP was lower than that of laserphyrin. Nevertheless, the effect on tumor progression after PLP PDT was virtually same as that after laserphyrin PDT (Figure 8). These results indicate the efficacy of PLP PDT for biliary tract cancer.

## 5. Conclusions

We demonstrated that the accumulation of PLP and PLP PDT effects are cancer cell-specific phenomena. PLP PDT was strongly cytotoxic at low concentrations compared to laserphyrin PDT. It is expected that photosensitivity could be attenuated. We propose, as one of the underlying mechanisms, that PLP is taken up by cells through HCP1. Moreover, PLP PDT inhibited tumor growth in NOZ tumor-bearing mice. We believe that PLP PDT will provide a survival advantage to patients with bile duct cancer.

## Figures and Tables

**Figure 1 ijms-22-07306-f001:**
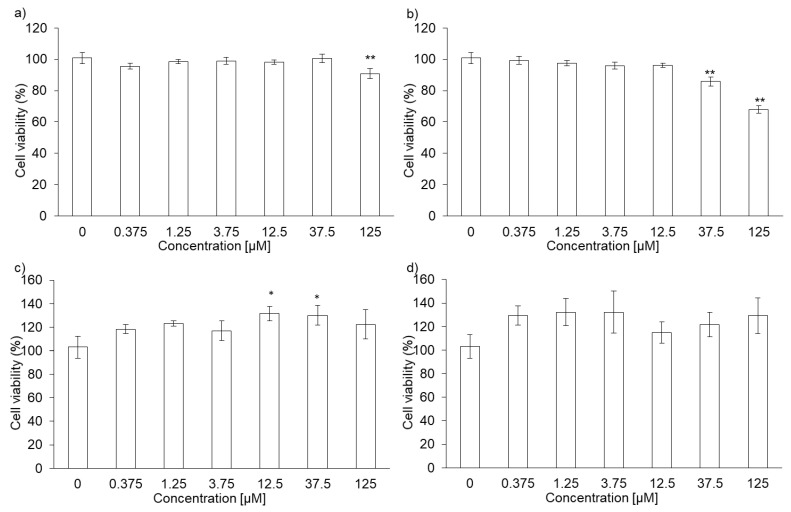
Cytotoxicity of PLP and laserphyrin. RGK1 (**a**) and RGM1 (**c**) cells were exposed to culture medium containing several concentrations of PLP. RGK1 (**b**) and RGM1 (**d**) cells were exposed to culture medium containing laserphyrin. Data are expressed as means ± SD (*n* = 5). * *p* < 0.05, ** *p* < 0.01.

**Figure 2 ijms-22-07306-f002:**
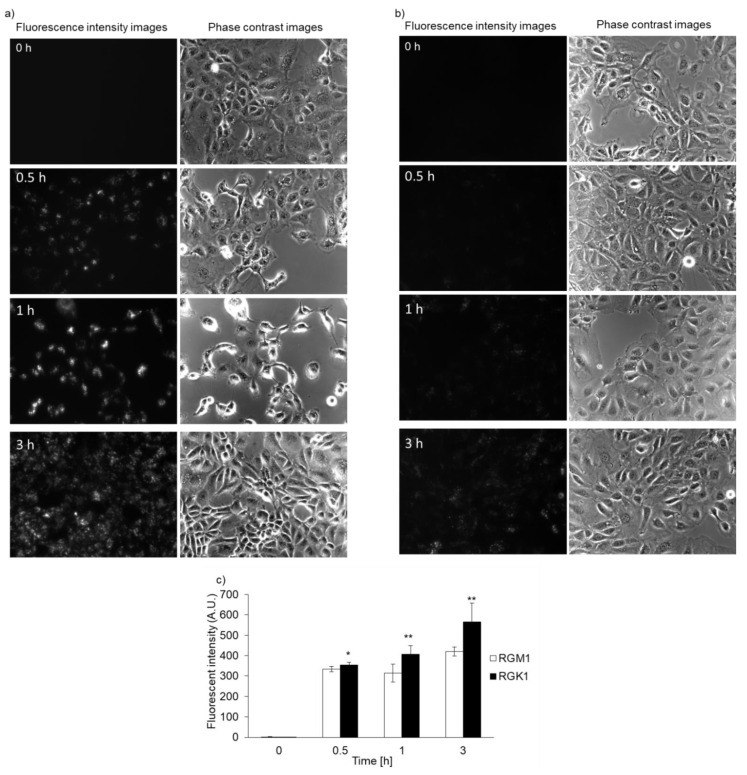
Intracellular PLP accumulation. Microscopy images of RGK1 (**a**) and RGM1 (**b**). Intracellular fluorescence intensity of PLP was measured using a microplate reader (**c**). Data are expressed as means ± SD (*n* = 5). * *p* < 0.05, ** *p* < 0.01.

**Figure 3 ijms-22-07306-f003:**
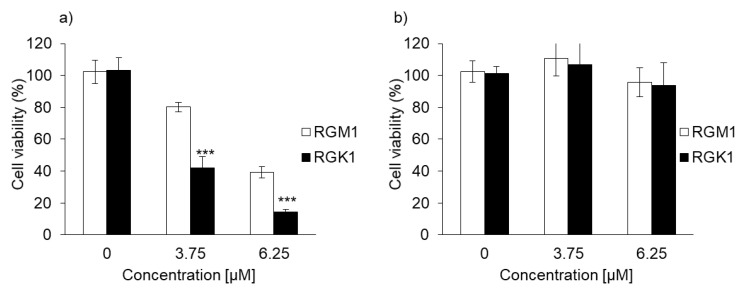
The cytotoxicity effect of PDT using (**a**) PLP and (**b**) laserphyrin. Data are expressed as means ± SD (*n* = 5). *** *p* < 0.001.

**Figure 4 ijms-22-07306-f004:**
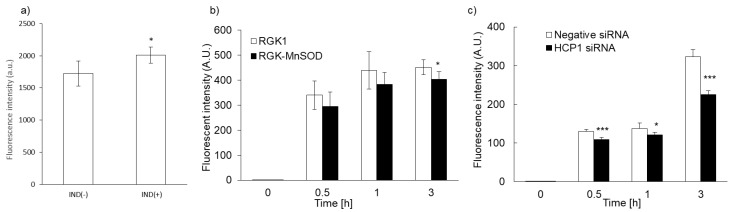
Relationship between PLP uptake and HCP1 expression. Cells were exposed to IND for 1 h (**a**). PLP accumulation in RGK1 was higher than in RGK-MnSOD (**b**). PLP accumulation was significantly decreased by HCP1 knockdown (**c**). Data are expressed as means ± SD (*n* = 5). * *p* < 0.05, *** *p* < 0.001.

**Figure 5 ijms-22-07306-f005:**
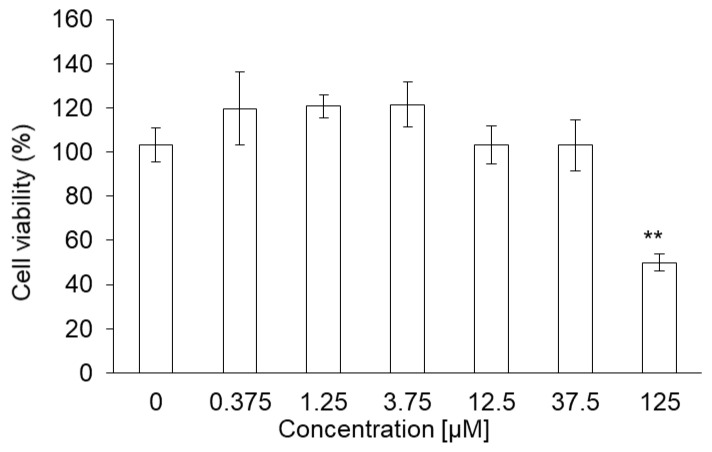
Cytotoxicity of PLP in NOZ cells. Data are expressed as means ± SD (*n* = 5). ** *p* < 0.01.

**Figure 6 ijms-22-07306-f006:**
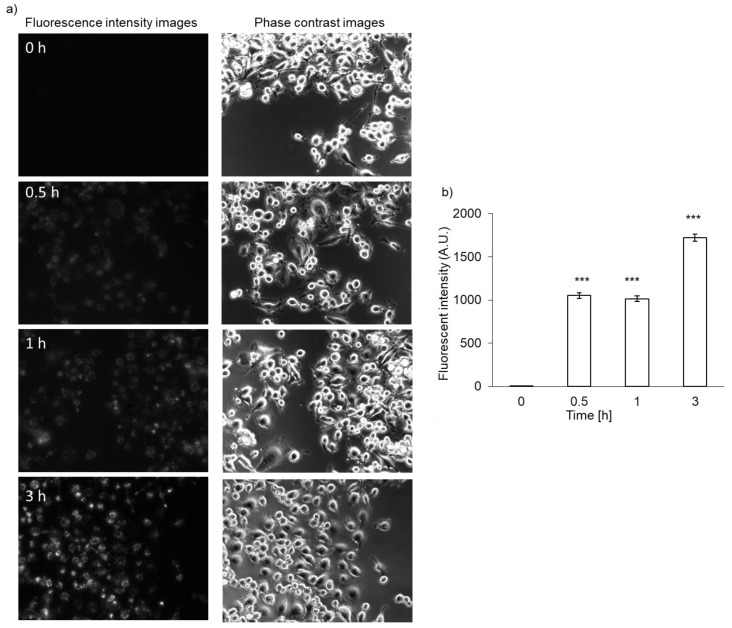
Intracellular PLP accumulation in NOZ cells. Microscopy images (**a**) and intracellular fluorescence intensity (**b**). Data are expressed as means ± SD (*n* = 5). *** *p* < 0.001.

**Figure 7 ijms-22-07306-f007:**
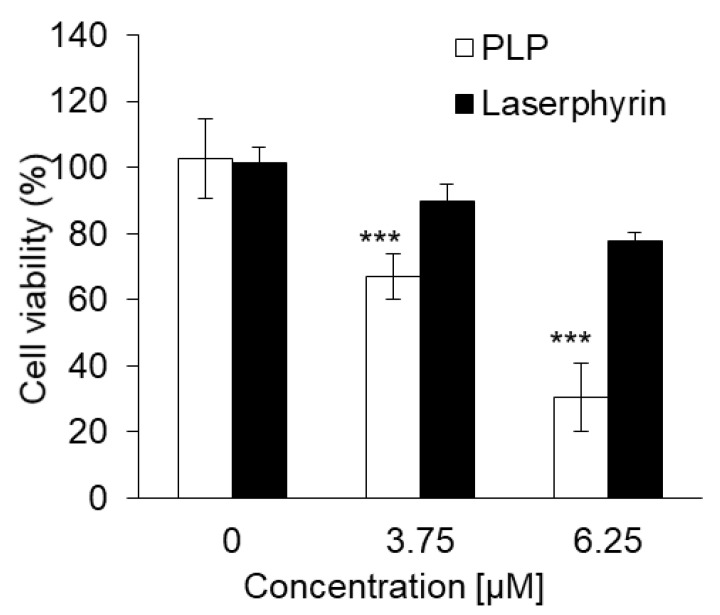
Cytotoxicity of PDT using PLP and laserphyrin in NOZ cells. Data are expressed as means ± SD (*n* = 5). *** *p* < 0.001.

**Figure 8 ijms-22-07306-f008:**
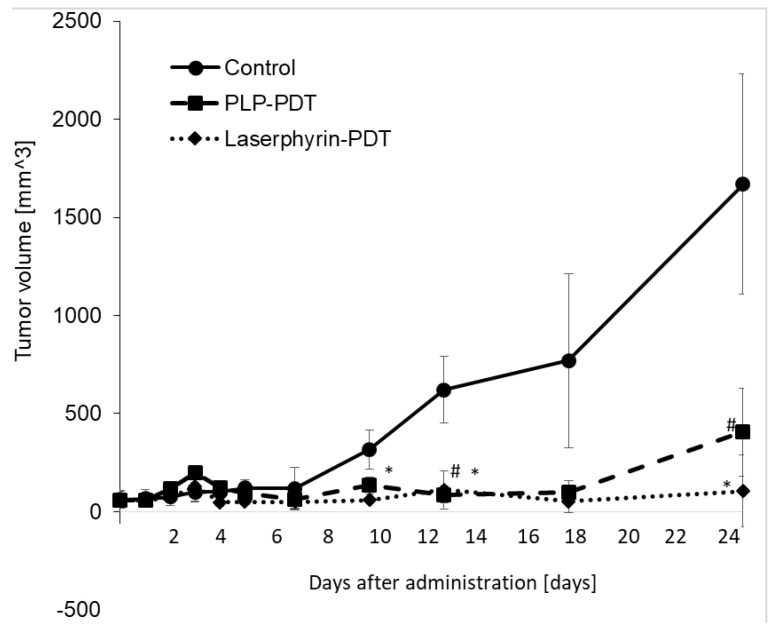
Cytotoxicity effect of PDT using PLP and laserphyrin in vivo. Data are expressed as means ± SD (*n* = 3). # *p* < 0.05 Control vs. PLP PDT. * *p* < 0.05 Control vs. laserphyrin PDT.

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
