# Peer review of "Porphylipoprotein Accumulation and Porphylipoprotein Photodynamic Therapy Effects Involving Cancer Cell-Specific Cytotoxicity"

_ijms, 2021, doi:10.3390/ijms22147306_

Round 1

Reviewer 1 Report

My comment is described in attached file.

Author Response

Major problem

  1. page 9, line 292-293: Authors describe “Considering the tumor volume in the control to be 100%, the volumes in PLP-PDT and Laserphyrin-PDT were 24% and 6%, respectively” Between PLP-PDT and Laserphyrin-PDT, is there significant difference?

→Tumor volumes in PLP-PDT and Laserphyrin-PDT did not show different significantly.

  1. Page 10-11: In discussion, there is no comment about Figure 8. Authors should discuss about Figure 8. In addition, in Figure 7, “the viability of cells subjected to PLP-PDT was significantly lower compared with that of cells subjected to Laserphyrin-PDT”. Authors also should comment about difference between Figures 7 and 8.

→According to reviewer’s comments, we revised sentences indicate red characters in page 12-13.

Miinor problem

  1. Page 5, line 204: T Cytotoxicity of PLP and Laserphyrinin cancer and normal cells.→ What is this?

→We mistake this sentence. We and cut it.

  1. Page 6, line 229: 79.0%±1.68%→79.0±1.7%
  2. Page 6, line 230: 40.5%±4.68%→40.5±4.7%
  3. Page 6, line 231: 38.4%±3.38%→38.4±3.4%, 14.0%±1.33%→14.0±1.3%
  4. Page 9, line 277: 65%±6.24%→65.0±6.2%
  5. Page 9, line 278: 88.9%±5.52%→88.9±5.5%
  6. Page 9, line 279: 26.5%±6.04%→26.5±6.0%, 77.5%±2.44%→77.5±2.4%

→According to reviewer’s comments, we revised sentences indicate red characters.

Reviewer 2 Report

Title: Reactive Oxygen species regulates the porphylipoprotein accumulation and photodynamic therapy using porphylipoprotein involves cancer specific cytotoxicity

Author: Kurokawa et al.

Comments: This manuscript by Kurokawa et al. is an interesting study, as it presents an approach to study the uptake of photosensitizer (PS; porphylipoprotein and laserphyrin) in rat normal gastric epithelial cells and its cancer-like mutant cells. The authors showed a significant increase in accumulation of PS and cytotoxic effects following PDT, specifically observed in cancer cells, relative to normal cells. Inhibitory effects of PLP-PDT and laserphyrin-PDT were also observed on the growth of NOZ (a gall bladder carcinoma cell line) tumors in murine model studies.

In my opinion, due to several deficiencies listed below, the paper could not be published in International Journal of Molecular Sciences in its present form.

  1. The title of the manuscript starts with reactive oxygen species (ROS) without any analysis or quantification of ROS in the entire manuscript.
  2. The title should have read as: “Reactive Oxygen species regulates the porphylipoprotein accumulation and porphylipoprotein-photodynamic therapy effects involving cancer cell-specific cytotoxicity”.
  3. Photosensitive disorder in lines 14, 15 and 75 probably means PS distribution or uptake, hence should be corrected.
  4. The mechanism of cancer-cell specific accumulation of PS is well known. For example, please visit the review by Castano et al. Photodiagnosis Photodyn Ther. 2005 June ; 2(2): 91–106. doi:10.1016/S1572-1000(05)00060-8, and modify the Introduction section.
  5. The sub-heading in results section 3.1 HT-induced mitochondrial ROS generation in cancer cells, is out of context and meaningless because neither mitochondria nor ROS were discussed in entire sub-section. Also, HT and WST assay should be spelled out.
  6. In Results section 3.8, the formula length x width2/2 for calculation of tumor volume is not understandable. Authors should clarify/correct.
  7. The results presented in Fig. 7 (in vitro) and Fig. 8 (in vivo) show opposite results. Laserphrin-PDT in cells seem to be inferior to PLP-PDT but appears to be more effective in murine tumor model studies. Authors should discuss the differences and explain the opposite results presented in two figures.
  8. An overall deficiency with respect to sample size, i.e. using n=4 for in vitro studies and n=3 for murine tumor model studies, respectively, was observed. Most of the scientific journals suggest using n=5 at least, for publication. The authors should consider adding more replicates to their experiments.  

Author Response

1. The title of the manuscript starts with reactive oxygen species (ROS) without any analysis or quantification of ROS in the entire manuscript.

→In this study, we did not perform ROS analysis, but we have already shown relationship between ROS and porphyrin uptake. So we added sentences indicate red characters in page 12.

2. The title should have read as: “Reactive Oxygen species regulates the porphylipoprotein accumulation and porphylipoprotein-photodynamic therapy effects involving cancer cell-specific cytotoxicity”.

→According to reviewer’s comments, we revised the title.

3. Photosensitive disorder in lines 14, 15 and 75 probably means PS distribution or uptake, hence should be corrected.

→Compared with conventional PS, PS can not release from PLP if nanoparticle of PLP does not disrupt. Thus, we consider that photosensitive disorder in PLP is lower than that in conventional PS.

4. The mechanism of cancer-cell specific accumulation of PS is well known. For example, please visit the review by Castano et al. Photodiagnosis Photodyn Ther. 2005 June ; 2(2): 91–106. doi:10.1016/S1572-1000(05)00060-8, and modify the Introduction section.

→According to reviewer’s comments, we revised sentences indicate red characters in page 1-2.

5. The sub-heading in results section 3.1 HT-induced mitochondrial ROS generation in cancer cells, is out of context and meaningless because neither mitochondria nor ROS were discussed in entire sub-section. Also, HT and WST assay should be spelled out.

→We missed the title of this result. According to reviewer’s comments, we revised sentences indicate red characters.

6. In Results section 3.8, the formula length x width2/2 for calculation of tumor volume is not understandable. Authors should clarify/correct.

→We revise formula and added reference [Euhus DM, Hudd C, LaRegina MC, Johnson FE. Tumor measurement in the nude mouse. J Surg Oncol. 1986;31:229–234. doi: 10.1002/jso.2930310402.] and [Tomayko MM, Reynolds CP. Determination of subcutaneous tumor size in athymic (nude) mice, Cancer Chemother. Pharmacol. 1989;24:148–154.]

7. The results presented in Fig. 7 (in vitro) and Fig. 8 (in vivo) show opposite results. Laserphrin-PDT in cells seem to be inferior to PLP-PDT but appears to be more effective in murine tumor model studies. Authors should discuss the differences and explain the opposite results presented in two figures.

→According to reviewer’s comments, we revised sentences indicate red characters in page 12-13.

8. An overall deficiency with respect to sample size, i.e. using n=4 for in vitro studies and n=3 for murine tumor model studies, respectively, was observed. Most of the scientific journals suggest using n=5 at least, for publication. The authors should consider adding more replicates to their experiments.

→According to reviewer’s comments, we change the sample size and revised Figure 1 to 7.

Round 2

Reviewer 2 Report

Reviewer’s comments:

My overall impression of the revised manuscript is, a job done in a hurry without carefully addressing the reviewers concerns and lack of proofreading before submitting this revised manuscript.

A few concerns are listed below (in italics):  

  1. The title of the manuscript starts with reactive oxygen species (ROS) without any analysis or quantification of ROS in the entire manuscript.

→In this study, we did not perform ROS analysis, but we have already shown relationship between ROS and porphyrin uptake. So we added sentences indicate red characters in page 12.

Since ROS was not analyzed in this research study, it can’t be used in the title. Its use in the title conveys a misleading information to the readership that ROS was analyzed in this research study. The scenario could have been different if it was a review article. The authors should remove the word ROS from the title.

  1. The mechanism of cancer-cell specific accumulation of PS is well known. For example, please visit the review by Castano et al.Photodiagnosis Photodyn Ther. 2005 June ; 2(2): 91–106. doi:10.1016/S1572-1000(05)00060-8, and modify the Introduction section.

→According to reviewer’s comments, we revised sentences indicate red characters in page 1-2.

This additional sentence was unnecessary and out of context. If authors do not wish to add the scientific evidence provided in the above-mentioned review, they should delete these two sentences (lines 43-45). The mechanisms of cancer cell-specific accumulation of PS are known with some details.   

  1. The sub-heading in results section 3.1 HT-induced mitochondrial ROS generation in cancer cells, is out of context and meaningless because neither mitochondria nor ROS were discussed in entire sub-section. Also, HT and WST assay should be spelled out.

→We missed the title of this result. According to reviewer’s comments, we revised sentences indicate red characters.

The first sentence of this section “T cytotoxicity of PLP and Laserphyrin in cancer and normal cells” seems out of context. Please delete this sentence.  

  1. An overall deficiency with respect to sample size, i.e. using n=4 for in vitro studies and n=3 for murine tumor model studies, respectively, was observed. Most of the scientific journals suggest using n=5 at least, for publication. The authors should consider adding more replicates to their experiments.

→According to reviewer’s comments, we change the sample size and revised Figure 1 to 7.

It is amazing that authors have added another replicate (n=4 to n=5) to their experiments in figures 1-7 in less than a week from rejection to resubmission of this version of the manuscript. It is surprising to see that figures 1-7 had no changes in the values except the scale on Y axis looks different in some cases (figures 2-5 and 7).

The values on Y axis for figure 4c are too different in two submissions, i.e. 0-400 AU vs. 0-10000 AU, respectively.

In figure 2c, *. ** and *** should be explained in legends.

In figure 5, ** and *p<0.05 are not matching.

In figure 6, number of samples and statistical test should be provided.

In figure 7, *** are shown with explanation for * and **. Please correct the legend.

  1. In discussion, please replace “occur” with “result in” in line 364.    

Author Response

A few concerns are listed below (in italics):  

  1. The title of the manuscript starts with reactive oxygen species (ROS) without any analysis or quantification of ROS in the entire manuscript.

→In this study, we did not perform ROS analysis, but we have already shown relationship between ROS and porphyrin uptake. So we added sentences indicate red characters in page 12.

Since ROS was not analyzed in this research study, it can’t be used in the title. Its use in the title conveys a misleading information to the readership that ROS was analyzed in this research study. The scenario could have been different if it was a review article. The authors should remove the word ROS from the title.

→According to reviewer’s comment, we revised the title to “The porphylipoprotein accumulation and porphylipopro-tein-photodynamic therapy effects involving cancer cell-specific cytotoxicity”

  1. The mechanism of cancer-cell specific accumulation of PS is well known. For example, please visit the review by Castano et al.Photodiagnosis Photodyn Ther. 2005 June ; 2(2): 91–106. doi:10.1016/S1572-1000(05)00060-8, and modify the Introduction section.

→According to reviewer’s comments, we revised sentences indicate red characters in page 1-2.

This additional sentence was unnecessary and out of context. If authors do not wish to add the scientific evidence provided in the above-mentioned review, they should delete these two sentences (lines 43-45). The mechanisms of cancer cell-specific accumulation of PS are known with some details.   

→According to reviewer’s comments, we revised sentences indicate red characters in page 1-2.

  1. The sub-heading in results section 3.1 HT-induced mitochondrial ROS generation in cancer cells, is out of context and meaningless because neither mitochondria nor ROS were discussed in entire sub-section. Also, HT and WST assay should be spelled out.

→We missed the title of this result. According to reviewer’s comments, we revised sentences indicate red characters.

The first sentence of this section “T cytotoxicity of PLP and Laserphyrin in cancer and normal cells” seems out of context. Please delete this sentence.  

→According to reviewer’s comments, we delete this sentence.

  1. An overall deficiency with respect to sample size, i.e. using n=4 for in vitro studies and n=3 for murine tumor model studies, respectively, was observed. Most of the scientific journals suggest using n=5 at least, for publication. The authors should consider adding more replicates to their experiments.

→According to reviewer’s comments, we change the sample size and revised Figure 1 to 7.

It is amazing that authors have added another replicate (n=4 to n=5) to their experiments in figures 1-7 in less than a week from rejection to resubmission of this version of the manuscript. It is surprising to see that figures 1-7 had no changes in the values except the scale on Y axis looks different in some cases (figures 2-5 and 7).

The values on Y axis for figure 4c are too different in two submissions, i.e. 0-400 AU vs. 0-10000 AU, respectively.

→We mistook the analysis of figure 4c. We normalized and attach the normalized figure 4c.

In figure 2c, *. ** and *** should be explained in legends.

In figure 5, ** and *p<0.05 are not matching.

In figure 6, number of samples and statistical test should be provided.

In figure 7, *** are shown with explanation for * and **. Please correct the legend.

→According to reviewer’s comments, we revised the legends.

  1. In discussion, please replace “occur” with “result in” in line 364.  

→According to reviewer’s comment, we replaced occur” with “result in”.